# Usability of a hearing test mobile app across generations

Małgorzata Pastucha[1,2]*, Elżbieta Gos[1,2], Krzysztof Kochanek[1,2], Henryk Skarżyński[1,2], Wieslaw Wiktor Jedrzejczak[1,2]

**1** Institute of Physiology and Pathology of Hearing, Warsaw, Poland, **2** World Hearing Center, Kajetany, Poland

* m.pastucha@ifps.org.pl

## Abstract

### Introduction

Traditional diagnostic methods of hearing assessment, such as pure tone audiometry, may not be equally accessible to everyone due to geographical or mobility limitations. Utilizing a mobile application (app) for self-assessment of hearing is a promising alternative. However, the effectiveness of apps, as well as their usability across different age groups, remains largely unexplored. The objective of the present study was to assess, across different age groups, the usability of the "Hearing Test" app which allows self-testing of hearing on a mobile phone.

### Materials and methods

The study was conducted on 77 participants from three age groups (16–39 years, 40–59 years, 60 years and older) who self-tested their hearing thresholds using the mobile app and who later underwent pure tone audiometry with an audiologist. The usability of the app was evaluated using a questionnaire based on the Mobile App Rating Scale (MARS), which was complemented by participant observation and interview.

### Results

The app generally yielded results comparable to pure tone audiometry. However, older age groups tended to report higher levels of difficulty across several usability dimensions. Specifically, the oldest group rated the app lower in terms of functionality (M = 2.30; SD = 1.27) and engagement-customization (M = 2.11; SD = 1.28). For the oldest participants, the greatest difficulties related to installation (48%), and interpretation of results (26%). None of the participants aged 60 or older were able to complete the test independently, in contrast to 67% of the youngest participants and 28% of the middle-aged who did not require assistance. All age groups expressed a

**Data availability statement:** The data underlying the results presented in the study are available in the Supporting information.

**Funding:** The author(s) received no specific funding for this work.

**Competing interests:** The authors have declared that no competing interests exist.

preference for a conventional hearing test over an app-based assessment, although the youngest group showed the greatest openness to using mobile apps.

## Conclusions

The "Hearing Test" app has demonstrated its potential as a tool for initial hearing assessment, particularly among younger users. However, older individuals often encounter difficulties with installation, interpretation of results, and overall usability. Adapting the interface to meet the specific needs of older users, including user-friendly tutorials and clear presentation of results, is crucial for enhancing its usability.

---

## Introduction

Hearing is one of the most important human senses, fostering linguistic and cognitive development and facilitating interpersonal communication [1]. Regrettably, various factors such as advancing age [2], prolonged exposure to loud noise [3], and exposure to harmful substances [4,5] can lead to hearing impairment. Very often people are unaware of the problem until it has reached a critical stage, at which point intervention may become more challenging and its effectiveness reduced. One potential solution to this problem is to implement regular hearing tests.

The gold standard for the diagnosis of hearing loss is pure tone audiometry (PTA). Unfortunately, access to hearing diagnostics is not equally available to all individuals. Those of advanced age or residing in rural areas are at a disadvantage, particularly if they have mobility issues [6]. The rapid development of new technologies in recent years, notably in the domain of teleaudiology, may potentially offer an effective solution to this issue.

Advances in telemedicine technology have streamlined diagnostic procedures, enabling more efficient and accurate healthcare delivery [7]. A growing interest in telehealth services has also led to a growth in audiology software and apps [8]. Currently, the most common sort of audiological apps are for testing hearing status, tinnitus management, and vertigo treatment [9]. A review of the literature shows that these tools are widely available and can be used by patients in the comfort of their own homes, with the added benefit of being low-cost or free [10]. Studies conducted on both adults and children have demonstrated that hearing test apps provide high reliability and good repeatability [11–14]. However, it should be emphasized that in a number of these studies, testing of hearing was conducted by an audiologist or other qualified individual, not the user themself [11,15]. Additionally, in these prior studies, participants did not evaluate the apps used. This creates a potential knowledge gap in that, as well as making the app reliable under laboratory conditions, it is also important that the app be reliable in the hands of the patient as well as being user-friendly [16].

In order to properly assess the usability of a mobile app designed for hearing testing, a clear understanding of terms is essential. According to ISO/IEC 9241−11 [17], usability is defined as "the extent to which a product can be used by specified users

to achieve specified goals with effectiveness, efficiency, and satisfaction in a specified context of use". This definition emphasises the importance of apps working effectively in real-world environments. This means avoiding highly technical or complex apps, no matter how advanced, which fail to meet the practical needs of everyday users. This understanding is essential given that factors such as age, education level, digital literacy, and health significantly affect how users perceive the usability of an app [18].

Health Working Papers published by the Organisation for Economic Co-operation and Development (OECD) in 2020 showed that an increasing number of people are using new technologies to monitor and control their health at home. However, there is a disparity in the level of use of these technologies depending on the age of the patient. The data indicate that about 40% of individuals between the ages of 55 and 74 used the internet for health information, in comparison to 61% of those between the ages of 25 and 54 [19]. Additionally, in the United States, individuals aged 65 and above were nearly 35 times less likely to utilize video consultations than those between the ages of 25 and 44 [20]. The observed differences can be attributed to the fact that today's seniors were born and raised at a time when computers and the internet were either rare or nonexistent [21]. Consequently, digital skills may present a significant barrier to the use of new technologies in healthcare. Furthermore, the use of apps can be challenging for seniors due to physical and manual limitations [21,22]. All these factors mean that older people may perceive the devices as less useful.

Age-related constraints are highly relevant, given the fact that life expectancy has increased globally, resulting in a demographic shift towards an older population. At the same time, the prevalence of chronic diseases has risen, calling for extra healthcare services to meet these new demands [23]. Ironically, then, the individuals who struggle the most to adapt to modern telemedicine solutions (i.e., seniors) are the ones who may benefit the most from this type of healthcare.

The rationale for the current study was to understand how users of different ages rated a mobile app for self-testing hearing thresholds, focusing on the different design and usability needs across age groups. Hearing ability and familiarity with technology tend to differ with age, so analysis of feedback from various demographics could reveal specific app modifications that might enhance accessibility, engagement, and overall user satisfaction. Exploring this aspect might lead to better hearing health outcomes. While some previous work has demonstrated the potential value of mobile apps in hearing screening, it has not focused on users' real-world experiences [11,15].

To address this gap, the aim of the current study was to evaluate the usability of the "Hearing Test" mobile application [24–27] for self-administered hearing assessments across different age groups. Specifically, the study focused on: (1) gauging participants perceived ease-of-use, (2) identifying the challenges users encountered in installing, conducting, and interpreting the test, and (3) examining participants' preferred hearing assessment methods.

## Methods

### Study procedure and data collection

The study included audiological assessments, a survey, observation, and interview. Audiological assessment was conducted both by the participant (self-testing of hearing with the app) and by the audiologist (pure tone audiometry).

Participants were recruited from the guests, employees, and their families of a sanatorium in Southern Poland. Data were collected by a single researcher during a structured, three-stage procedure.

In the first stage, each participant received written instructions (in Supplementary Material) on how to perform a hearing test using the "Hearing Test" mobile app for Android devices [24–27]. The instructions outlined steps including preparation for the test, installing the app, conducting the test, and sending the results to an email address. If the participant did not have an Android smartphone or bundled headphones (i.e., the headphones included in the set provided by the manufacturer of the smartphone), the researcher provided these tools. Throughout this stage, the researcher observed the participants to assess their ability to complete the task independently. If difficulties arose, the researcher intervened and documented the issue on an observation sheet. After the test, the results were sent by the participant to the provided email address. The researcher then discussed the results with the participant, focusing on their ability to interpret the

audiogram and understand key parameters such as intensity and frequency of sound. The participant was also asked to assess the clarity and usefulness of the results and whether it indicated normal hearing or a hearing problem.

The second stage was a hearing screening performed by an audiologist (the researcher). The participant was instructed by the audiologist on how to respond during the test by pressing a button upon hearing a tone. After the test, the audiologist explained the results to the participant and provided recommendations.

In the final stage, participants completed a paper-and-pencil survey.

Both tests were conducted in the same quiet room, and all procedures were done during one visit. The procedures are outlined in Fig 1.

### Audiological assessment

**Mobile app.** The "Hearing Test" app was used in this study [25–27]. It was chosen based on the following criteria: free to use, available in Polish, and with the ability to determine hearing thresholds. Further details regarding selection of this app are available in Pastucha et al. [28].

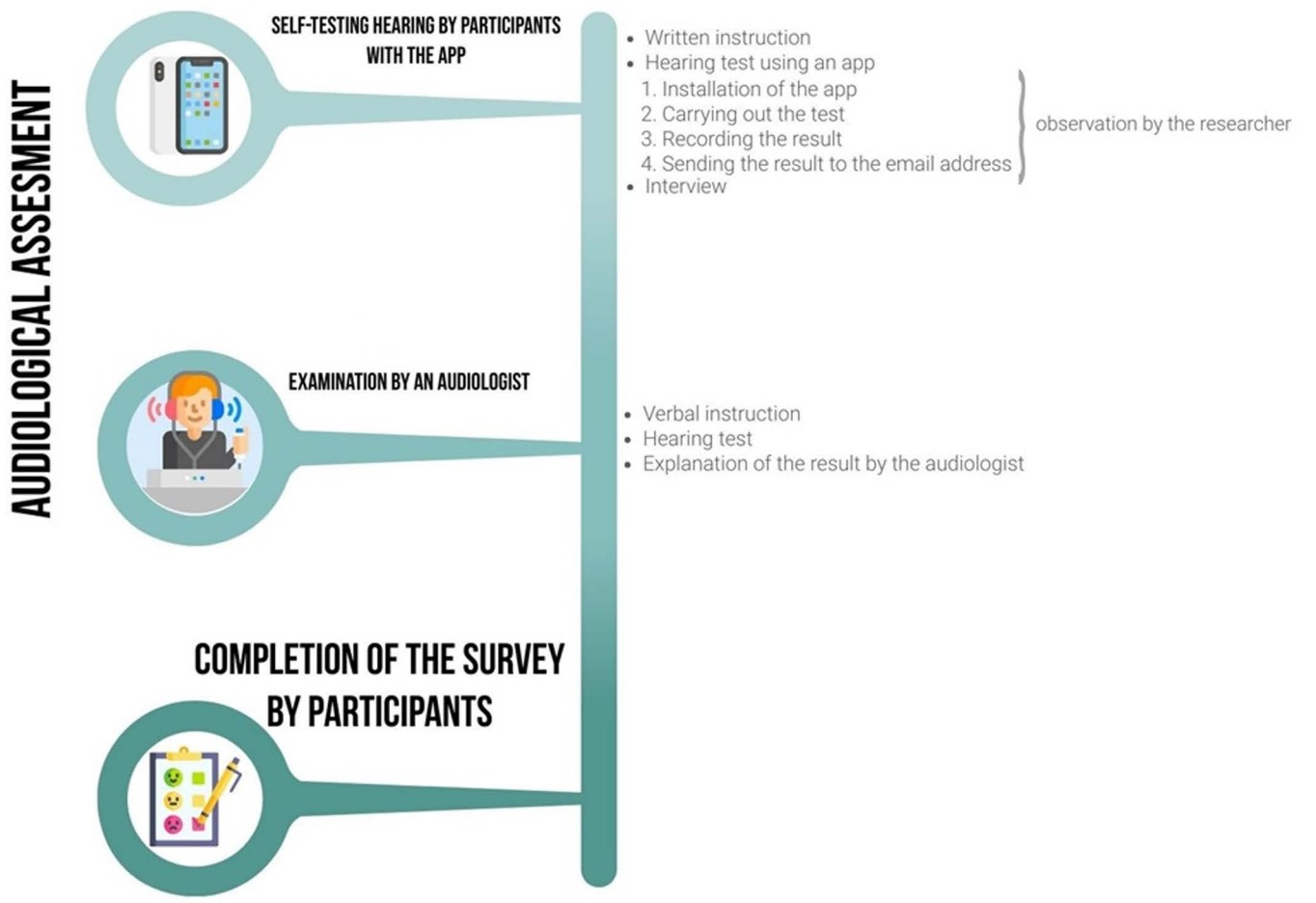

**Fig 1. Study procedure.**

The app can determine hearing thresholds in each ear at the following frequencies: 250, 500, 1000, 2000, 4000, 6000, and 8000 Hz (standard settings). The levels range from −10–100 dB HL (decibel hearing level) using pulsed sounds. The hearing threshold is assessed using a descending method in 5 dB steps, where the subject presses buttons labeled "I can hear", "I can't hear", and "at the hearing limit".

Results are presented as an audiogram, showing degrees of hearing loss for both ears using standard audiological symbols. The app averages hearing thresholds at 500, 1000, 2000, and 4000 Hz for each ear and classifies the degree of hearing loss according to WHO guidelines. Based on the results, it also lists potential symptoms and further recommendations. Additionally, users can customize the presentation of the audiogram to show degree of hearing loss, age norms, or relationships to the typical frequency and sound levels of human speech.

The app offers several customization options, including headphone selection and calibration, choice of test frequencies, intensity settings, and masking conditions. Binaural testing is also available. Users can save and compare results with previous tests and monitor background noise during testing.

Although the "Hearing Test" app was originally designed to assess air conduction only, its developers have recently introduced a version capable of bone conduction testing using a dedicated bone vibrator [29]. However, this hardware is not yet commercially available to the general public and is currently used primarily in research and professional settings. For this reason, and in order to ensure accessibility and scalability, we selected the air-conduction-only version of the app, which remains freely available and does not require additional specialized equipment.

**Pure tone audiometry.** Pure tone audiometry (air conduction only) was performed by an audiologist using the Sense Examination Platform [30]. Sennheiser HDA 200 headphones were utilized so as to provide effective acoustic isolation [31]. Air conduction was assessed separately for each ear at frequencies of 250, 500, 750, 1000, 1500, 2000, 3000, 4000, 6000, and 8000 Hz, following the modified Hughson–Westlake procedure [32]. Sound levels ranged from 0 to 80 dB HL, with thresholds determined by obtaining two out of three responses at each level.

**Survey.** A survey designed for the study consisted of a metrics section and a main section divided into two parts.

Part I contained questions about rating the app. The questions were based on the Mobile App Rating Scale (MARS) [33]. The survey evaluates the usability of the app across four dimensions:

- engagement (both interest and customization),
- functionality,
- aesthetics,
- information (both quality and quantity of information).

Each element was evaluated using 5 answering options (described in the Supplementary Material- Table 1). For the purpose of analysis, these answers were assigned a score on a scale from 1 (first response) to 5 (last response).

Part II, which is the researcher's original contribution, addressed other aspects of usability:

- difficulties in using the app,
- understanding the results provided,
- preferred method of hearing assessment.

The complete list of utilized questions can be found in Supplementary Material.

## Observation and interview

The researcher observed the participants while they self-tested their hearing using the app. Observation focused on the participant's independence in carrying out the test. Following the test, the researcher interviewed the participant to assess

their ability to correctly interpret the results displayed by the app. Findings were recorded on an observation sheet (in Supplementary Material).

## Ethical considerations

Participants were prospectively recruited between 5 August 2023 and 15 October 2023. Prior to participation, all subjects were informed about the nature and aims of the study. Written informed consent was obtained from all participants. In the case of minors aged 16–18 years, written consent was also obtained from their parents or legal guardians. The study was conducted in accordance with the guidelines of the Declaration of Helsinki and was approved by the Ethics Committee of the Institute of Physiology and Pathology of Hearing (consent no. KB.IFPS 3/2023).

**Table 1. Socio-demographic characteristics of the participants.**

| | ≤39 years (n=21) | 40–59 years (n=29) | ≥60 years (n=27) |
|---|---|---|---|
| **Age (years)** | | | |
| Range | 16–39 | 43–59 | 60–88 |
| M (SD) | 28.9 (8.4) | 49.7 (5.2) | 73.1 (7.2) |
| **Sex, *n* (%)** | | | |
| Women | 14 (66.7) | 19 (65.5) | 19 (70.4) |
| Men | 7 (33.3) | 10 (34.5) | 8 (29.6) |
| **Education, *n* (%)** | | | |
| Primary | 7 (33.3) | 1 (3.4) | 4 (14.8) |
| Vocational | 1 (4.8) | 8 (27.6) | 5 (18.5) |
| Secondary | 5 (23.8) | 14 (48.3) | 11 (40.7) |
| Tertiary | 8 (38.1) | 6 (20.7) | 7 (25.9) |
| **Employment, *n* (%)** | | | |
| Student | 6 (28.6) | – | |
| Professionally active | 12 (57.1) | 24 (82.8) | 3 (11.1) |
| Unemployed | 3 (14.3) | 2 (6.9) | – |
| Pensioner | – | 3 (10.3) | 24 (88.9) |
| **Financial situation, *n* (%)** | | | |
| Bad/very bad | 1 (4.8) | – | 2 (7.4) |
| Fair | 3 (14.3) | 15 (51.7) | 13 (48.1) |
| Good/very good | 17 (81.0) | 14 (44.4) | 12 (44.4) |
| **Experience with hearing examinations, *n* (%)** | | | |
| First examination | 14 (66.7) | 16 (55.2) | 12 (44.4) |
| Several years ago | 2 (9.5) | 11 (37.9) | 12 (44.4) |
| Regularly | 5 (23.8) | 2 (6.9) | 3 (11.1) |
| **Hearing aid, *n* (%)** | | | |
| No | 20 (95.2) | 29 (100.0) | 25 (92.6) |
| Yes | 1 (4.8) | – | 2 (7.4) |
| **Frequency of using mobile devices, *n* (%)** | | | |
| Every day | 21 (100.0) | 24 (82.8) | 11 (40.7) |
| Several times a week | – | 1 (3.4) | 6 (22.2) |
| Several times a month | – | 2 (6.9) | – |
| Less than once a month | – | – | 2 (7.4) |
| Do not use | – | 2 (6.9) | 8 (29.6) |

## Participants

The study group comprised individuals who had expressed an interest in participating. Inclusion criteria were: 1) age ≥ 16 years, and 2) ability to read and speak Polish. Exclusion criteria were: 1) lack of free time, 2) refusal to install the app on their own device; 3) insufficient cooperation. All participants were in good health, physically fit, and of comparable physical and mental condition.

Initially, the study group comprised 80 subjects. However, the results from three participants were deemed unreliable and subsequently excluded. Therefore, the final study group consisted of 77 subjects, comprising 52 women and 25 men, aged between 16 and 88 years, with a mean age of 52.2 years (SD = 18.8). The participants were divided into three age groups: ≤ 39 years (21 subjects), 40–59 years (29 subjects), and ≥ 60 years (27 subjects).

## Statistical analysis

The socio-demographic characteristics were examined using descriptive statistics and percentages. The assumption of normality was checked with a Shapiro–Wilk test. Differences between age groups were assessed using a one-way ANOVA with post-hoc Bonferroni test and a χ2 test for independence.. A p-value <0.05 was considered statistically significant. Analyses were conducted using IBM SPSS Statistics (version 24).

## Results

### Characteristics of the participants

The socio-demographic characteristics of the three groups are set out in Table 1. It shows that the proportions of men and women in all three groups were similar. The youngest participants were relatively the most educated, while the middle-aged were the most professionally active. The reported financial situation of the two oldest groups was slightly worse compared to the youngest participants. For the majority of the youngest and middle-aged participants, the hearing examination was the first they had had, while the oldest participants had had more experience with hearing tests. Older participants were less likely to declare daily use of a mobile device than the youngest subjects. This trend is particularly evident in the 60 + group, where 40.7% of participants reported using mobile devices daily, while 29.6% said they did not use a mobile device at all.

### Results of self-testing with the app and with pure tone audiometry

Table 2 shows the results of hearing examinations conducted using the app (self-testing) and pure tone audiometry (clinical testing), employing the World Health Organization (WHO) guidelines to classify the degree of hearing loss [34].

As can be seen in Table 2, the results obtained via the app and the clinical test (PTA) are similar. For the youngest group, the app overestimated the percentage of normal hearing. Conversely, in the two older groups, the app more frequently indicated hearing loss than was indicated by pure tone audiometry. This pattern is particularly noticeable for the 60 + group, especially in the left ear – where PTA indicated 25% of subjects had normal hearing, while the app showed only 11%.

### Results of assessment by the mobile app

Participants of different ages rated specific aspects of the application differently. In five out of the six assessed dimensions, statistically significant differences were observed in terms of age. The most pronounced effect (F = 32.08; p < 0.001; e2 = 0.46) was noted in the Functionality dimension (ease of use): participants aged 60 and over (M = 2.30; SD = 1.27) rated this aspect significantly lower than either the youngest group (M = 4.00; SD = 0.80) or the middle-age group (M = 4.33; SD = 0.73).

**Table 2. Percentages of individuals with normal hearing and hearing loss (HL) according to self-testing with the app and with pure tone audiometry (PTA).**

| Hearing status | ≤39 years (n=21) App | PTA | 40–59 years (n=29) App | PTA | ≥60 years (n=27) App | PTA |
|---|---|---|---|---|---|---|
| **Right ear, %** | | | | | | |
| Normal | 90.5 | 85.7 | 72.4 | 79.3 | 14.8 | 18.5 |
| Mild HL | 4.8 | 4.8 | 20.7 | 13.8 | 55.6 | 44.4 |
| Moderate HL | 4.8 | 4.8 | 3.4 | 3.4 | 18.5 | 18.5 |
| Moderately severe HL | – | 4.8 | – | – | 7.4 | 14.8 |
| Severe HL | – | – | – | – | 3.7 | 3.7 |
| Profound HL | – | – | 3.4 | 3.4 | – | – |
| **Left ear, %** | | | | | | |
| Normal | 100 | 90.5 | 58.6 | 75.9 | 11.1 | 25.0 |
| Mild HL | – | 4.8 | 31.0 | 13.8 | 44.4 | 25.9 |
| Moderate HL | – | 4.8 | 6.9 | 6.9 | 25.9 | 29.6 |
| Moderately severe HL | – | – | 3.4 | 3.4 | 14.8 | 11.1 |
| Severe HL | – | – | – | – | 3.7 | 7.4 |
| Profound HL | – | – | – | – | – | – |

Age differences were also observed in in the aspect Engagement–customization (F=8.63; p<0.001; e2=0.19). Again, the participants aged ≥60 (M=2.11; SD=1.28) rated this aspect significantly lower than both the youngest participants (M=3.52; SD=1.47) and the middle-aged participants (M=3.34; SD=1.26).

The same was true for the aspect Information–quality (F=5.42; p=0.006; e2=0.13). Participants aged 60 and over (M=2.81; SD=1.59) rated this aspect significantly lower than both the youngest participants (M=3.95; SD=1.20) and the middle-aged participants (M=3.83; SD=1.23).

Similar age-related differences were also observed in the aspect Information–quantity (F=4.96; p=0.010; e2=0.12). Again, the participants aged 60 and over (M=2.74; SD=1.43) rated this aspect significantly lower than both the youngest participants (M=3.76; SD=1.18) and the middle-aged participants (M=3.72; SD=1.33).

Age differences were also observed in the aspect Aesthetics (F=3.48; p=0.036; e2=0.09). The participants aged 60 and over (M=3.22; SD=0.75) rated this aspect significantly lower than the middle-aged participants (M=3.79; SD=0.86) and similar to the youngest participants (M=3.67; SD=0.91).

Only in one dimension, Engagement–interest, were the ratings similar across age groups (F=0.96; p=0.388). The mean rating in the youngest group was M=4.24 (SD=0.94); in the middle-aged group M=4.24 (SD=0.94); and in the oldest group M=3.85 (SD=0.91).

Mean ratings are shown in Fig 2.

## Difficulties in using the app

After completing the hearing test, participants were asked which element of the app was the most difficult for them, and they had to select one option from a list: installing the app, device calibration, performing the test, saving the test results, and interpretation of the results. For each age group, the number of participants selecting a specific option was counted, and the percentage was calculated for that age group. The results are shown in Fig 3. The oldest participants most often reported problems with installing the app, with 48.1% finding this task difficult, compared to only 4.8% of the youngest participants. For the youngest group, the most difficult aspect was interpreting the test results provided by the app, which 57.1% of them found difficult. Similarly, 48.3% of middle-aged participants had difficulty with this task, compared to 25.9%

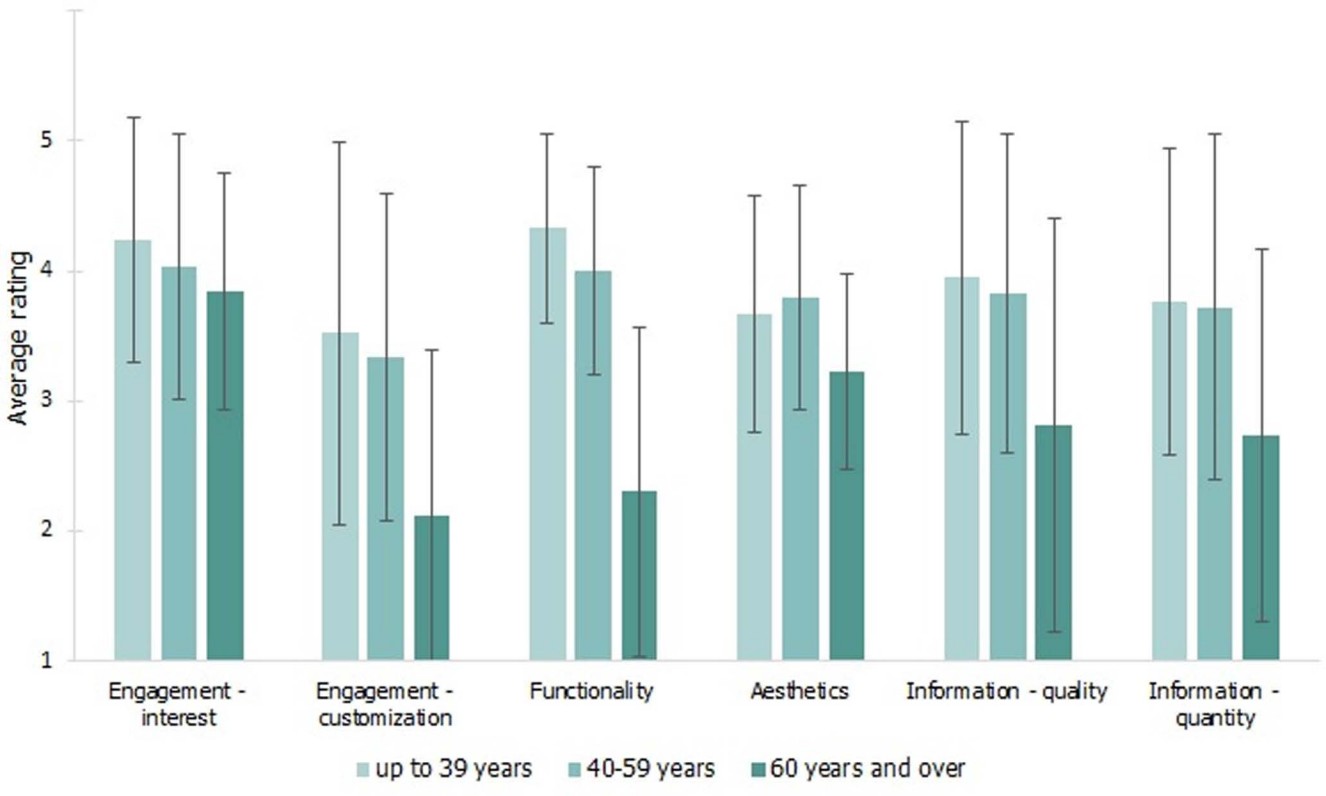

**Fig 2. Assessment of the mobile app by subjects of different ages.** (Means and SDs).

of the oldest group. However, the differences between age groups did not reach statistical significance: $\chi2(8) = 14.8$; p = 0.063.

The researcher observed participants throughout the procedure. The majority of the youngest participants (66.7%) completed the examination independently (without the researcher's assistance), while 33.3% required partial assistance, such as help with specific tasks like installing the app or saving results. In the middle-aged group, 27.6% of participants performed the examination alone, 69.0% required partial assistance, and 3.4% needed full assistance. In the 60+group, all participants required the researcher's assistance, with 66.7% needing partial assistance and 33.3% requiring full assistance, in which the researcher performed all steps, including interacting with the smartphone (Nevertheless, all of them still answered the question about what was the most difficult aspect of performing the test.). Differences between the groups were statistically significant ($\chi^2(4) = 34.88$, p < 0.001).

### Understanding the results provided by the app

Most participants in the youngest group (52.4%), the middle-aged group (48.3%), and the oldest group (51.9%) indicated that the written description was the easiest way to understand the test results. The audiogram (graph) was less preferred, with respective preferences of 28.6%, 31.0%, and 25.9%. Likewise, the calculated degree of hearing loss was less favored, with preferences of just 19.0%, 20.7%, and 22.2%, respectively. The differences between the age groups were not statistically significant ($\chi^2(4) = 0.24$, p = 0.993).

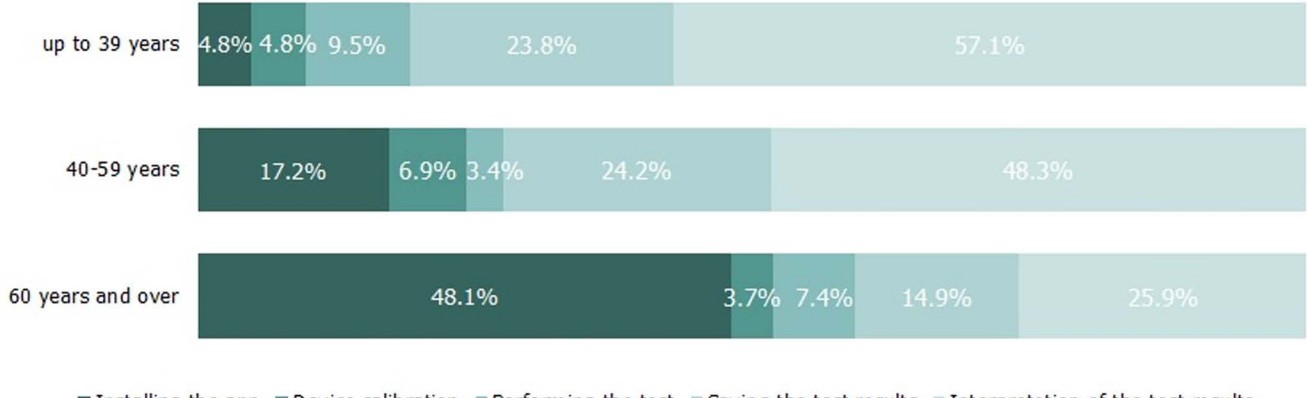

**Fig 3. Percentage distribution of difficulties encountered in using the app within each age group.** Within each age group percentages add up to 100%.

Participants found it challenging to interpret the app's hearing test results. The participants' interpretations were checked by the researcher. The percentage of correct interpretations was the highest in the youngest group (38.1%), lower in the middle-aged group (17.2%), and lowest in the older group (3.7%). Moreover, the percentage of totally incorrect interpretations was the highest in the older group (44.4%), lower in the middle-aged group (27.6%), and lowest in the youngest group (14.3%). The differences between the groups were statistically significant ($\chi2(4) = 11.52$; $p = 0.021$).

### Preferred method of hearing assessment

Most participants in the youngest group (61.9%), the middle-aged group (79.3%), and the oldest group (77.8%) preferred a conventional diagnostic hearing test as their favoured method of hearing assessment. Using the app was less favoured, with respective preferences of 28.6%, 17.2%, and 18.5%. No preference was reported by 9.5%, 3.4%, and 3.7% of individuals, respectively. Differences between the age groups were not statistically significant ($\chi^2(4) = 2.48$, $p = 0.649$).

However, in one particular aspect a significant difference between the age groups emerged. The youngest participants preferred using a mobile app at their primary care physician's office (57.1%) rather than visiting an audiology clinic for a diagnostic hearing test (42.9%). In the middle-aged group, the preferred option was to go to an audiology clinic for a diagnostic hearing test (65.5%) rather than using the app at a general practitioner's (GP) (34.5%). Similarly, in the oldest group, there were more supporters of a conventional hearing test at an audiology clinic (77.8%) than using the app at the GP's (22.2%). The differences between the age groups were statistically significant ($\chi^2(2) = 6.30$, $p = 0.043$).

### Discussion

The aim of this study was to assess the usability of a mobile application for hearing testing across diverse age groups. The findings indicate that the app was capable of producing outcomes comparable to those obtained by the conventional method employed by professionals in clinical settings. Nevertheless, some discrepancies were observed, depending on the age of the participant. This suggests that older users may encounter more challenges with the app, and point to potential usability concerns within this group.

A comparison of hearing test results obtained via the app and the conventional method showed that the app generally provided similar results to professional hearing tests, especially in the youngest group, where concordance of results was highest. However, the app showed a tendency to overestimate hearing loss in the other two (older) age groups. The

differences may be due to a better understanding of the technology and greater proficiency in its use by younger people [21], or perhaps to the difficulty of using the app or the lower accuracy of self-testing by older users [18,35,36].

It is also possible that tinnitus may have interfered with test performance in some participants, especially among older adults. Tinnitus is known to complicate tone detection, particularly during self-assessments, by masking or mimicking test stimuli. Although we did not specifically assess the presence of tinnitus in our participants, its higher prevalence among older adults may partly explain some of the discrepancies observed between app-based and conventional audiometry results in this age group [37,38].

We also investigated how perceptions of the app differed according to age. In only one dimension – engagement–interest – were there consistent ratings across all age groups. This measure indicates that the app has universal appeal. At the same time, significant variations were noted in the other five aspects: functionality, engagement–customization, aesthetics, information–quality, and information–quantity. In terms of functionality, the oldest group exhibited the greatest variability in their responses compared to the other groups. This tendency was not seen in evaluations of the other aspects, where all groups demonstrated similar levels of response variability. This effect may be due to the broader diversity of the oldest group, which could include both technologically experienced individuals and those without experience. In contrast, the younger groups might be more homogeneous in this regard, potentially affecting how they assessed functionality while having less impact on their evaluation of other aspects.

Older users encountered difficulties with functionality and customization, rating these aspects lower than did younger and middle-aged individuals. Also, our study found that these usability challenges often contributed to a reduction in self-efficacy when using the app in this age group. This problem underscores the need for a more intuitive interface for older adults. Our study highlights that there were significant age-related differences in app usability and satisfaction, suggesting that older adults may have different expectations and requirements when using health-related apps. In contrast to our findings, Tao and Or [39] found that, in other healthcare contexts, older adults showed high acceptance and satisfaction with apps. However, it should be noted that this research only considered involved users over the age of 55, which limits the comparability of results with other age groups and this may account for some of the differences in perceptions. Finally, Tao and Or used a different type of app, one specifically designed to track easily recorded measures like weight, temperature, blood pressure, and blood glucose levels.

Our study focused on hearing, which requires a significantly different approach compared to apps designed for medication monitoring or health management. Rather than requiring a relatively straightforward task such as entry of data, a hearing test requires the user to carefully detect sounds of various frequencies and provide an accurate response. The test procedure includes specific instructions, such as ignoring masking noise in the non-test ear; however, some participants, particularly those unfamiliar with pure tone audiometry, may have overlooked these instructions, displayed on the smartphone screen. This complexity may lead to lower user satisfaction. This might be particularly true among older people who may find it more difficult to intuitively work out the app and hence may rate it less favourably.

In terms of evaluating the app and being satisfied with it, the essential point is whether it does the job effectively. The more complex and intricate, the more important it is for the user to have confidence and familiarity. This has been confirmed by a recent study by Masiero and colleagues [40], who compared assessments made by patients and carers who used an app designed to monitor the patient's health status and support palliative care. They found that all participants rated the app similarly. The greatest discrepancy was found in users assessment in terms of aesthetics, with older patients rating the app better in this area. In contrast, the quantity and quality of information were rated similarly by both groups. These findings indicate that users are satisfied with the range and presentation of information. This suggests that, in the context of easy apps, the user's skills and extent of knowledge in a given health topic do not matter. This is corroborated by another study by Masiero on women with breast cancer [41], which found that 39% of participants rated the information in the app to be evidence-based, relevant, and trustworthy. It is important to note that the app studied by Masiero was relatively simple in its design. It consisted of three main sections: an educational module, a pain and psychological

well-being assessment module, and an e-diary, all of which allowed for close monitoring of a patient's physical status. In comparison, our study involved a more sophisticated hearing test app, and our findings were that the oldest age group was not particularly satisfied with the feedback they received in the test results. This may, of course, be due to the conventional "upside-down" way in which pure tone audiometry results are presented, which is tailored to professionals in the field and is difficult for the general public to understand.

Our findings are in accordance with those of other m-health studies, which have demonstrated that older individuals frequently encounter greater difficulties when using mobile health apps. A major difficulty our subjects encountered was installation [42]. Over 48% of the 60 + group faced challenges in installing the app. The test itself was also a significant issue. Seniors were often unable to complete the test independently due to inappropriate reaction to sounds (e.g., reacting to disruptive noises). This was compounded by inadequate button pressing and holding (e.g., pressing the return button), influenced by factors such as motor coordination, vision, and familiarity with the technology [18,35,43]

The end result was that in the 60 + age group, all participants required the assistance of the researcher, with 67% requiring partial assistance and 33% requiring full assistance. In contrast, 28% of the middle-aged group were able to perform the test independently, in comparison to the youngest group, who mostly performed the test without additional support.

While the greatest challenge in the oldest age group was installing the app, for the youngest participants the most difficult task was interpreting the results. Nevertheless, despite the difficulties encountered in this area, younger participants had a higher rate of accurate interpretation compared to older ones. Irrespective of age, however, the majority of individuals preferred descriptive hearing test results, with audiograms and calculated degrees of hearing loss rated as less comprehendible. Unfortunately, to the best of our knowledge, no other studies have yet been conducted to assess interpretation skill and preferred format of audiological test results, so we are unable to quote the findings of other researchers on this matter.

It should also be mentioned that some participants performed the test on a device provided by the researcher. Most participants who used the prepared smartphone owned a device with an iOS operating system, which was incompatible with the tested app. Nevertheless, these same individuals, had no difficulties installing or using the app on the provided Android smartphone. A second group of participants using the prepared smartphone were older adults who required full assistance in performing the hearing test and were totally unfamiliar with smartphone technology. We therefore assume that they would have needed assistance regardless of whether they were using their own device or the one provided.

Our results indicate that individuals across all age groups expressed a preference for undergoing a conventional hearing test in an audiology clinic, rather than using an app. At the same time, however, we found that younger users would prefer to go to their GP for an app-based test rather than a visit to a local audiology clinic. This preference should be set alongside findings from other studies where participants have generally expressed a preference for mobile-based tests over conventional audiometry [44,45]. Children have also been shown to prefer assessments using apps [46]. In our study, the youngest group showed a greater willingness to use the app at a GP's than did older participants. The differences may be due to the fact that the main concerns of younger people are performing the test correctly and interpreting the results correctly. If the app is operated by a specialist who also interprets the results, they show a greater willingness to use it. They feel more comfortable handing over control to a specialist, which gives them added confidence and a sense of being cared for by an expert.

## Limitation

A potential limitation of this study is the selection of an app that has a non-standard response option "at the hearing limit". This feature is intrinsic to the app's interface and not modifiable by researchers. However prior validation studies have demonstrated high agreement between the app's results and those obtained using traditional audiometric methods. This provides confidence in the reliability of the method, despite its deviation from traditional protocols. However, further

research is recommended to explore how this additional response option might influence results across different populations and environments.

An additional limitation of our study may be the wording of the question regarding the difficulties encountered during the test. Participants were only allowed to select one answer, with no option to indicate no problems or indicate multiple difficulties. This wording may have introduced ambiguity, especially for those who required full assistance to complete the test.

Another limitation is the lack of bone conduction assessment in both the app and the reference test. Consequently, we were unable to determine the type of hearing loss (i.e., conductive, sensorineural, or mixed), which limits clinical interpretation. However, this limitation is inherent to the nature of mobile hearing applications. Mobile hearing test apps are generally similar to screening audiometry rather than comprehensive air and bone conduction testing. These apps are designed for self-testing and provide preliminary hearing information that encourages users to seek full clinical evaluation rather than replace professional testing. We would not expect users to perform bone conduction self-testing, especially since even trained audiologists sometimes encounter difficulties with bone conduction testing in complex cases.

## Practical implications

In light of our findings, and from the conclusions of previous reports [47], several recommendations can be made to enhance the usability and effectiveness of apps for testing hearing:

1. Optimising the user interface

- Simplifying the interface, increasing the size and spacing of interactive elements, and potentially offering different or adaptive interfaces tailored to various age groups.

- Reducing the number of button presses to reduce the risk of error and user frustration.

- Devising ways to prevent accidental exits.

2. Detailed instructions

- Provide more detailed and clear instructions on how to perform the test, preferably in the form of a video rather than in writing.

3. Educational resources

- Creating a clearer system for explaining test results which can be understood by users of different ages and knowledge levels.

- Adding educational resources such as short instructional videos or infographics to help users better understand the test results and relevance to their health.

- Implementation of a contextual support function offering real-time assistance while using the app, for example in the form of a chatbot or virtual assistant.

## Conclusions

Mobile apps can be an effective tool for initial hearing assessment, especially among younger users. However, older participants faced challenges, including difficulties with installation, understanding instructions, and interpreting results. These findings highlight the need for simplified interfaces, user-friendly tutorials, and clear guidance to improve usability, particularly for older people.

Participants across all age groups expressed a preference for descriptive result formats over an audiogram. Younger users showed greater willingness to use the app in a clinical settings under professional supervision, reflecting greater confidence when they were guided by a specialist.

Despite these challenges, the app shows potential as an initial hearing assessment tool, provided it is adapted to meet the diverse needs and technical competencies of different age groups. Improving accessibility, ease of installation, and the clarity of result presentation would enhance user confidence and expand the app's utility in both clinical and non-clinical settings. These refinements are particularly important to support older users, ensuring consistent and reliable results across all demographics.

## Supporting information

**S1 File. Observation sheet, written instruction, complete list of utilized questions.**
(DOCX)

**S2 File. Anonymized data.**
(XLSX)

## Acknowledgments

The authors thank Andrew Bell for comments on an earlier version of this article. This article includes a modified graphic template (Fig 1) based on a free resource provided on www.freepik.com.

## Author contributions

**Conceptualization:** Małgorzata Pastucha, Elżbieta Gos, Krzysztof Kochanek, Henryk Skarżyński, Wieslaw Wiktor Jedrzejczak.

**Data curation:** Małgorzata Pastucha.

**Formal analysis:** Elżbieta Gos.

**Investigation:** Małgorzata Pastucha, Elżbieta Gos, Krzysztof Kochanek, Henryk Skarżyński, Wieslaw Wiktor Jedrzejczak.

**Methodology:** Małgorzata Pastucha, Elżbieta Gos, Wieslaw Wiktor Jedrzejczak.

**Supervision:** Krzysztof Kochanek, Henryk Skarżyński, Wieslaw Wiktor Jedrzejczak.

**Writing – original draft:** Małgorzata Pastucha, Elżbieta Gos, Wieslaw Wiktor Jedrzejczak.

**Writing – review & editing:** Małgorzata Pastucha, Elżbieta Gos, Wieslaw Wiktor Jedrzejczak.

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
