## [Decision Letter · Decision Letter 0]

PONE-D-25-24910Usability of a Hearing Test Mobile App Across GenerationsPLOS ONE

Dear Dr. Pastucha,

Thank you for submitting your manuscript to PLOS ONE. After careful consideration, we feel that it has merit but does not fully meet PLOS ONE’s publication criteria as it currently stands. Therefore, we invite you to submit a revised version of the manuscript that addresses the points raised during the review process.

**ACADEMIC EDITOR: ** **Smart phone apps for hearing evaluation became popular during the COVID 19 pandemic. However, they continue to have several limitations. Few test bone conduction hearing and most are not user friendly especially for older patients.  The apps can be useful for surveillance or monitoring. ** **Authors have stated that "** The app generally yielded results comparable to pure tone audiometry". Did the app test bone conduction? Table 2 mentions severity of hearing loss but does not mention whether the loss was conductive, sensorineural of mixed. How were the App vs Audiometry test results in patients with asymmetrical conductive or sensorineural hearing loss?  Please discuss. Also, please respond to Reviewer 1's comments.

We look forward to receiving your revised manuscript.

Kind regards,

Gauri Mankekar, MD,PhD,FACS

Academic Editor

PLOS ONE

Journal Requirements:

2. In the online submission form, you indicated that “The data underlying the results presented in the study are available from the corresponding author upon reasonable request.”

Reviewers' comments:

Reviewer's Responses to Questions

**Comments to the Author**

1. Is the manuscript technically sound, and do the data support the conclusions?

Reviewer #1: Yes

Reviewer #2: Yes

2. Has the statistical analysis been performed appropriately and rigorously? 

Reviewer #1: I Don't Know

Reviewer #2: Yes

3. Have the authors made all data underlying the findings in their manuscript fully available?

Reviewer #1: Yes

Reviewer #2: No

4. Is the manuscript presented in an intelligible fashion and written in standard English?

Reviewer #1: Yes

Reviewer #2: Yes

5. Review Comments to the Author

Reviewer #1: Regarding the dissimilarity of results obtained using the app compared with conventional audiometry, consider that hearing loss will tend to worsen with age, and tinnitus often accompanies that hearing loss, making self-assessment more challenging (i.e. the patient will confuse the test tone with the tinnitus). You could therefore conceivably attribute the progressively less similar results as the participants age to tinnitus rather than incompetence with technology.

Reviewer #2: The study provides valuable insight into usability of mobile apps for hearing across generations and is of interest to the telemedicine, audiology, and app development communities. The authors state that the data are available "from the corresponding author upon reasonable request". It would be good if anonymized datasets could be made available.

6. PLOS authors have the option to publish the peer review history of their article (what does this mean? ). If published, this will include your full peer review and any attached files.

**Do you want your identity to be public for this peer review?** For information about this choice, including consent withdrawal, please see our Privacy Policy .

Reviewer #1: **Yes: ** Cynthia Collier

Reviewer #2: No

---

## [Author Response · Author response to Decision Letter 1]

18 Jun 2025

Dear Editor,

We would like to thank you for your valuable feedback and comments on our manuscript. Your suggestions have greatly helped us to improve the clarity and quality of the article. Below are our responses to your comments:

Smart phone apps for hearing evaluation became popular during the COVID 19 pandemic. However, they continue to have several limitations. Few test bone conduction hearing and most are not user friendly especially for older patients. The apps can be useful for surveillance or monitoring.

Authors have stated that "The app generally yielded results comparable to pure tone audiometry". Did the app test bone conduction? Table 2 mentions severity of hearing loss but does not mention whether the loss was conductive, sensorineural of mixed. How were the App vs Audiometry test results in patients with asymmetrical conductive or sensorineural hearing loss? Please discuss.

Response:

1. Bone conduction testing

The mobile app used in our study assessed air conduction thresholds only. No bone conduction testing was performed, either by the app or performed by an audiologist. Mobile hearing test apps are generally similar to screening audiometry rather than comprehensive air and bone conduction testing. These apps are designed for self-testing and provide preliminary hearing information that encourages users to seek full clinical evaluation rather than replace professional testing. We would not expect users to perform bone conduction self-testing, especially since even trained audiologists sometimes encounter difficulties with bone conduction testing in complex cases.

In the manuscript, we used the term pure tone audiometry (PTA) as a commonly understood shorthand for air conduction audiometry conducted by an audiologist. We acknowledge that this terminology may be misleading, as PTA formally includes both air and bone conduction measurements. We have clarified this point in the revised manuscript to prevent any ambiguity.

We have added following sentences: “Although the “Hearing Test” app was originally designed to assess air conduction only, its developers have recently introduced a version capable of bone conduction testing using a dedicated bone vibrator [29]. However, this hardware is not yet commercially available to the general public and is currently used primarily in research and professional settings. For this reason, and in order to ensure accessibility and scalability, we selected the air-conduction-only version of the app, which remains freely available and does not require additional specialized equipment.” in lines 120-124.

And “Pure tone audiometry (air conduction only) was performed by an audiologist using the Sense Examination Platform” in line 126.

2. Lack of information on hearing loss type (conductive/sensorineural/mixed)

Since the app and the audiological examination both measured only air conduction thresholds, we were unable to determine the type of hearing loss (i.e., whether it was conductive, sensorineural, or mixed). The classification in Table 2 refers exclusively to the degree of hearing loss based on air conduction thresholds, following WHO guidelines.

We have clarified this in the limitation section: “Another limitation is the lack of bone conduction assessment in both the app and the reference test. Consequently, we were unable to determine the type of hearing loss (i.e., conductive, sensorineural, or mixed), which limits clinical interpretation. However, this limitation is inherent to the nature of mobile hearing applications. Mobile hearing test apps are generally similar to screening audiometry rather than comprehensive air and bone conduction testing. These apps are designed for self-testing and provide preliminary hearing information that encourages users to seek full clinical evaluation rather than replace professional testing. We would not expect users to perform bone conduction self-testing, especially since even trained audiologists sometimes encounter difficulties with bone conduction testing in complex cases.” in lines 363-369.

3. Performance in asymmetrical hearing loss

The app includes an automatic masking feature, which is activated when stimuli are presented at high intensity levels. In such cases, masking noise is delivered in the non-test ear, accompanied by an on-screen instruction informing the user not to respond to the noise. This feature is intended to reduce the risk of cross-hearing and improve measurement accuracy in cases of asymmetrical hearing. A description of this function is provided in lines 117–118 and 298–301 of the manuscript.

We also sincerely thank the reviewers for their insightful and constructive comments. Below we provide our detailed responses to each point raised, along with the corresponding changes made to the manuscript.

Reviewer #1: Regarding the dissimilarity of results obtained using the app compared with conventional audiometry, consider that hearing loss will tend to worsen with age, and tinnitus often accompanies that hearing loss, making self-assessment more challenging (i.e. the patient will confuse the test tone with the tinnitus). You could therefore conceivably attribute the progressively less similar results as the participants age to tinnitus rather than incompetence with technology.

Response:

We thank the reviewer for this comment. Indeed, tinnitus is commonly associated with age-related hearing loss and can interfere with tone perception during hearing tests. However, based on direct observation of participants during the app-based testing, we noted that older adults most frequently struggled with technical aspects of the procedure, including app installation, interpreting the interface, and following the testing instructions. These observations suggest that usability challenges played a more prominent role than auditory interference in this context. However, we recognize that tinnitus may have additionally impacted some participants’ ability to detect test tones, particularly in self-administered conditions.

We have included a sentence in the Discussion section: “It is also possible that tinnitus may have interfered with test performance in some participants, especially among older adults. Tinnitus is known to complicate tone detection, particularly during self-assessments, by masking or mimicking test stimuli. Although we did not specifically assess the presence of tinnitus in our participants, its higher prevalence among older adults may partly explain some of the discrepancies observed between app-based and conventional audiometry results in this age group [37,38].” in lines 272-276.

Reviewer #2: The study provides valuable insight into usability of mobile apps for hearing across generations and is of interest to the telemedicine, audiology, and app development communities. The authors state that the data are available "from the corresponding author upon reasonable request". It would be good if anonymized datasets could be made available.

Response:

We thank the reviewer for the comment. We have prepared the relevant anonymized data and included them as supplementary material with this revised submission.

---

## [Editor Report · Decision Letter 1]

Usability of a Hearing Test Mobile App Across Generations

PONE-D-25-24910R1

Dear Dr. Pastucha,

We’re pleased to inform you that your manuscript has been judged scientifically suitable for publication and will be formally accepted for publication once it meets all outstanding technical requirements.

Kind regards,

Gauri Mankekar, MD,PhD,FACS

Academic Editor

PLOS ONE
---

## [Editor Report · Acceptance letter]

PONE-D-25-24910R1

PLOS ONE

Dear Dr. Pastucha,

I'm pleased to inform you that your manuscript has been deemed suitable for publication in PLOS ONE. Congratulations! Your manuscript is now being handed over to our production team.

Kind regards,

on behalf of

Dr. Gauri Mankekar

Academic Editor

PLOS ONE